# Sequential Endoluminal Gemcitabine and Cabazitaxel with Intravenous Pembrolizumab as a Bladder-Preserving Strategy for Docetaxel-Unresponsive Non-Muscle Invasive Urothelial Carcinoma Following Transurethral Resection of Bladder Tumor

**DOI:** 10.3390/cancers16142561

**Published:** 2024-07-17

**Authors:** Ian M. McElree, Vignesh T. Packiam, Ryan L. Steinberg, Helen Y. Hougen, Sarah L. Mott, Mohamad Abou Chakra, Yousef Zakharia, Michael A. O’Donnell

**Affiliations:** 1Carver College of Medicine, University of Iowa, Iowa City, IA 52242, USA; ian-mcelree@uiowa.edu; 2Rutgers Cancer Institute of New Jersey, Division of Urology, Department of Surgery, New Brunswick, NJ 08903, USA; vignesh.packiam@rutgers.edu; 3Department of Urology, University of Iowa, Iowa City, IA 52242, USA; ryan-steinberg@uiowa.edu (R.L.S.); helen-hougen@uiowa.edu (H.Y.H.); mohamad-abouchakra@uiowa.edu (M.A.C.); 4Holden Comprehensive Cancer Center, University of Iowa, Iowa City, IA 52242, USA; sarah-mott@uiowa.edu (S.L.M.); yousef-zakharia@uiowa.edu (Y.Z.); 5Division of Hematology, Oncology, Blood & Marrow Transplantation, Department of Internal Medicine, University of Iowa, Iowa City, IA 52242, USA

**Keywords:** pembrolizumab, urinary bladder neoplasms, cabazitaxel, gemcitabine

## Abstract

**Simple Summary:**

After failing the first-line treatment of high-risk non-muscle invasive urothelial carcinoma (NMIUC), salvage intravesical therapies are important to preserve patient quality of life and avoid radical surgery. However, some patients will develop a disease resistant to commonly used drug regimens. We aimed to evaluate a novel combinatory therapy consisting of intravesical sequential gemcitabine and cabazitaxel with concomitant systemic pembrolizumab (GCP) for patients with recurrent high-risk NMIUC of the bladder and upper urinary tracts. We found in a population of 26 patients with 31 treated units (bladder and/or upper urinary tracts) that, following GCP treatment, over half of the treated units remained disease-free at 2 years. In this highly pretreated population, the progression of disease was a concern with an estimated 30% rate of progression at 2 years. However, only 4% of the population experienced cancer-related death during this same time. Lastly, this regimen was well tolerated, with all patients completing the scheduled induction course.

**Abstract:**

Growing evidence suggests that many patients with high-risk non-muscle invasive urothelial carcinoma (NMIUC) can undergo bladder-sparing management with salvage intravesical therapies. However, inherent or developed disease resistance, particularly after multiple lines of prior salvage therapy, implores the continued pursuit of new treatment combinations. Herein, we describe the outcomes of 26 patients (31 treated units; 24 lower tract, 7 upper tract) with high-risk NMIUC treated with sequential intravesical gemcitabine and cabazitaxel with concomitant intravenous pembrolizumab (GCP) at the University of Iowa from August 2020 to February 2023. Median (IQR) follow-up was 30 (IQR: 17–35) months. Treated units had a history of high-risk NMIUC with a median of four prior endoluminal inductions. Overall, 87% of units presented with CIS or positive urine cytology. The 1- and 2-year recurrence-free survival was 77% (CI: 58–88%) and 52% (CI: 30–70%), respectively. The 2-year progression-free and cancer-specific survival was 70% (CI: 44–85%) and 96% (CI: 75–99%), respectively. In total, 22/26 (85%) patients reported any adverse event and 5/26 (19%) reported a grade ≥3 adverse event; however, all patients tolerated a full induction course. These results suggest that GCP is an effective and tolerable treatment option for patients with recurrent high-risk NMIUC.

## 1. Introduction

High-risk non-muscle invasive urothelial carcinoma (NMIUC) of the bladder is treated with intravesical therapy, traditionally Bacillus Calmette–Guérin (BCG), following the transurethral resection of bladder tumor (TURBT) [1]. Unfortunately, approximately 40% of patients will recur following treatment with adequate intravesical BCG and subsequently be designated as BCG-unresponsive [2]. The current guidelines recommend offering these patients radical cystectomy given an increased likelihood of disease progression [1]. However, growing evidence suggests patients will benefit from further treatment with intravesical combinatory chemotherapy regimens, particularly gemcitabine and docetaxel (Gem/Doce) [3,4,5,6,7,8]. Unfortunately, resistance to these therapies may exist or develop.

While there is extensive research towards identifying predictors of response to BCG, the mechanisms of failure to Gem/Doce remain less clear. Multidrug resistance (MDR)-associated efflux pumps are a known contributor to treatment failure in cancer chemotherapy. The expression and up-regulation of MDR proteins in urothelial tissue following treatment with taxane chemotherapies have been demonstrated [9]. As extrapolated from other cancer types, the propensity for bladder cancer relapse likely increases following multiple lines of therapy with agents susceptible to MDR mechanisms [10]. Cabazitaxel is a taxane with reduced efflux pump affinity and sustained efficacy in the setting of docetaxel-resistant prostate cancer [11]. In addition to reapplying topical therapy, pembrolizumab, a systemic anti PD-1 immunotherapy, was FDA approved for BCG-unresponsive CIS and provides an additional means of cancer control to areas excluded from endoluminal drug delivery, such as the upper tract (UT) and prostatic urethra [12]. We report outcomes of patients treated with sequential intravesical gemcitabine and cabazitaxel (Gem/Cabaz) with concurrent intravenous pembrolizumab (GCP) for primarily (88%) BCG-bladder failure patients with high-grade (HG) docetaxel-unresponsive NMIUC who refused or were poor candidates for surgery, some of whom concurrently or metachronously developed the disease in extravesical sites.

## 2. Materials and Methods

### 2.1. Study Design and Population

After obtaining IRB approval, we retrospectively reviewed all patients with NMIUC between August 2020 and February 2023. Patients were included if intending to receive 6 weekly endoluminal instillations of sequential gemcitabine and cabazitaxel with concurrent systemic pembrolizumab therapy. Prior to induction, patients with visible tumor received complete resection or ablation. High-risk status was defined by American Urological Association criteria [1,13]. Patients were excluded from efficacy analysis if they did not have pathological high-grade disease or a definitively positive localizing high-grade cytology immediately prior to GCP induction. Additionally, patients were excluded from the analysis if they had not yet and/or did not undergo follow-up surveillance.

### 2.2. Gemcitabine/Cabazitaxel/Pembrolizumab Therapy

The intravesical Gem/Cabaz treatment protocol was derived from that of other sequential intravesical treatments, which have been previously reported [3,14]. In brief, patients were sequentially treated with 1 g of gemcitabine in 50 mL of normal saline for 90 min followed by 5 mg of cabazitaxel dissolved in 50 mL of normal saline for 90 min. When receiving treatment for the upper tract, each treatment drug was hung 30–35 cm above the flank level and set at a constant gravity rate of 1 drip every 3–4 s for a total of 90 min through a previously placed percutaneous nephrostomy tube. Patients were instructed to refrain from urinating for 60–120 min following Gem/Cabaz instillation. Patients were treated with 1300 mg oral sodium bicarbonate the evening prior and the morning of treatment to alkalinize the urine.

Induction Gem/Cabaz treatments were scheduled once a week for 6 weeks. Pembrolizumab therapy was initiated during the third Gem/Cabaz treatment and continued every 3 weeks until 5 treatments had been completed by 3 months, to permit evaluation of a full pembrolizumab treatment cycle. Subsequent Gem/Cabaz therapy evolved from monthly to every 3 weeks to coincide with pembrolizumab therapy and reduce patient visits. Thus, during the first 3 months, there were 6 once-weekly Gem/Cabaz induction treatments followed by 3 triweekly Gem/Cabaz treatments for a total of 9 instillations. Once a 3-month evaluation was clear of disease, GCP treatments were changed to every 6 weeks for up to 2 years until unacceptable toxicities or progression occurred. Maintenance Gem/Cabaz installation procedures and dosages matched those used in the induction protocol, but the pembrolizumab dose was doubled from 200 to 400 mg during the 6-week cycles [12,15]. Patients received routine laboratory monitoring and screening for adverse events (AEs) with each 6-week treatment.

### 2.3. Surveillance

Surveillance for cancer recurrence took place 3 to 5 weeks after ending the 3-month induction period and typically involved formal restaging under anesthesia. Formal restaging procedures included cystoscopy with blue-light, bladder barbotage cytology, bilateral upper tract barbotage cytologies, bilateral retrograde pyelograms, random bladder biopsies, and prostatic urethral biopsies. Routine UT investigations (CT, MRI, RGs, URS) were obtained every 6 months for the first 2 years with URS annually, at minimum. If disease-free, repeat surveillance cystoscopy with bladder cytology and FISH was performed quarterly for 2 years, and biannually afterwards.

### 2.4. Analysis

Data including patient clinicopathologic features, treatment history, AEs, and oncologic outcomes were retrospectively reviewed and analyzed. The primary outcome was HG recurrence-free survival (HGRFS). Recurrence was defined as HG tumor relapse in the bladder/upper tract or prostatic urethra in males. Secondary outcomes included duration of response (DOR), progression-free survival (PFS), cancer-specific survival (CSS), and overall survival (OS). AEs were abstracted from patient charts and classified by National Cancer Institute Common Terminology Criteria for Adverse Events (CTCAE) version 5. Survival probabilities were plotted using the Kaplan–Meier method. Estimates along with 95% pointwise confidence intervals were reported. HGRFS was defined as time from initiation of Gem/Cabaz induction to recurrence. Among patients without recurrence at the initial surveillance, DOR was defined as time from initial surveillance to recurrence. PFS was defined as the time from the earliest initiation of Gem/Cabaz induction to progression defined as T2+ disease, nodal or distant metastasis, or death due to cancer. Otherwise, patients were censored at last disease evaluation. CSS and OS were defined as time from the earliest initiation of Gem/Cabaz induction to death due to cancer or any cause, respectively. Patients still alive were censored at last follow-up. All statistical testing was two-sided and assessed for significance at the 5% level using SAS v9.4 (SAS Institute, Cary, NC, USA).

## 3. Results

### 3.1. Clinicopathological Characteristics

In total, 26 patients (31 treated units; 24 lower urinary tract, 7 upper urinary tract) with HG NMIUC were included in the final analysis (Table 1). An additional seven upper tract units were treated with GCP but did not meet the eligibility criteria and thus were excluded from the final efficacy analysis (Appendix A) but included in the toxicity evaluation.

Within the final analysis, 18 (58%) units presented with biopsy-proven CIS of the bladder (bCIS) or prostate (pCIS), and another 9 (29%) presented with presumed CIS in the setting of positive HG cytology. Of the treated lower urinary tracts, 8 (33%) had a history of pCIS, and 5 (21%) presented with pCIS immediately prior to GCP induction. The median number of prior inductions was 4, and all 24 (100%) units had been previously exposed to taxane-containing regimens with a median of 39 prior endoluminal docetaxel instillations.

Among all treated units, the median (IQR) number of Gem/Cabaz maintenance treatments was 7 (3–13; Table 2). One patient relapsed at their first surveillance and was ineligible for maintenance treatments. The median number of pembrolizumab treatments was 11 and 10 for units receiving therapy to the lower and upper tracts, respectively. Logistics related to delayed insurance approval and local treatment coordination led to two patients not receiving pembrolizumab therapy.

### 3.2. Tolerance

In total, 22 (85%) patients reported AEs following treatment (Table 3). There were a total of 67 AEs reported throughout treatment, of which 52 were attributed to Gem/Cabaz and 15 were attributed to pembrolizumab. The majority of AEs were mild and transient; 29 were grade 1, 33 were grade 2, and 5 were grade 3. Grade 3 AEs included the development of ciprofloxacin-resistant UTI, upper tract stone formation, pneumatosis intestinalis, capillary leak syndrome, and immune checkpoint inhibitor pancreatitis. All patients were able to tolerate a full induction course, though 6 (23%) patients ultimately declined further treatment secondary to side effects. There were no deaths associated with GCP treatment.

### 3.3. High-Grade Recurrence-Free Survival

Of the patients presenting with bladder disease, there were 12 HG recurrences during follow-up including 8 CIS, 2 HG cytology, 1 nodal metastasis noted on CT, and 1 muscle invasive disease upon cystectomy (Figure 1). Locations of CIS recurrences included 5/8 within the bladder, 2/8 within the prostatic ducts, and 1/8 within the prostatic urethra. Of those with prior pCIS, 4/8 experienced disease recurrence. There were two upper tract recurrences including one TaHG and one T1HG + CIS at nephroureterectomy. Among all units, complete response (CR) at first surveillance was 87%, and HGRFS was 77% (CI: 58–88%) and 52% (CI: 30–70%) at 1 and 2 years, respectively (Figure 2). Subgroup analysis of the lower urinary tracts demonstrated a CR of 83% and an HGRFS of 70% (CI: 47–85%) and 50% (CI: 26–71%) at 1 and 2 years, respectively (Appendix A). Of those disease-free at first surveillance, the median duration of response was 29 months (Appendix A).

### 3.4. Progression and Survival

Median follow-up for survival was 30 (IQR: 17–35) months. During the study period, six patients were elected to proceed with radical cystectomy and two with radical nephroureterectomy (Appendix A). In total, 5/6 patients undergoing cystectomy had pCIS, either prior to surgery or at time of cystectomy. One patient in the lower urinary tract treatment group developed HG cytology of an upper tract during cancer surveillance and elected for radical nephroureterectomy with final pathology pT2N0. First progression events included the development of metastasis in three patients (one distant, two node-positive at time of cystectomy), muscle invasive bladder cancer in two patients (one with node-positive disease), and urothelial cancer-related death in one patient. Aside from the patient experiencing a cancer-related death, all patients experiencing disease progression ultimately developed metastasis, three local and two distant. Of those with local metastasis, one patient received neoadjuvant gemcitabine and cisplatin followed by radical cystectomy, one was treated with adjuvant nivolumab, and one with T2N1 disease after cystectomy refused adjuvant nivolumab due to arthralgias experienced with prior pembrolizumab therapy. All three of these patients were disease-free at the conclusion of the study period. Of the two patients with distant metastasis, one was currently receiving enfortumab–vedotin with pembrolizumab and the other had received enfortumab–vedotin alone with no evidence of disease at the conclusion of the study period. Estimated PFS at 2 years was 70% (CI: 44–85%; Figure 3). Overall survival was 91% (CI: 67–98%) at 2 years (Appendix A). Cancer-specific survival was 96% (CI: 75–99%) at 2 years.

### 3.5. Excluded Upper Tract Units Receiving GCP Treatment

Of the additional seven upper tract units not meeting the study inclusion criteria, three had previously received upper tract treatment with docetaxel-containing regimens (Appendix A). Criteria for treatment included suspicious or atypical cytology ±>50% abnormal fluorescence in situ hybridization (FISH) results. At a median follow-up of 20 months, 5/7 patients remained recurrence-free. One patient experienced definitive recurrence with positive HG cytology at 35 months and another experienced non-definitive recurrence with 25/25 abnormal FISH results at 27 months.

## 4. Discussion

Our study has several important findings. First, in a heavily pretreated cohort with a median of 4 prior treatments and 27 prior docetaxel instillations among all units, the regimen was reasonably well tolerated with a total of five grade 3 AEs (19%). However, the ability to continue maintenance therapy was impacted by AEs with 6/26 (23%) patients stopping treatment. Among all units, there was a durable treatment response with an 87% CR and 52% 2-year RFS, which is notable given the extensive treatment history. Progression events were limited with 70% and 96% PFS and CSS at 2 years, respectively. Importantly, the regimen appears to have a distinctly favorable efficacy profile relative to single-agent pembrolizumab and may offer a viable method to overcome resistance to docetaxel-containing regimens, which are being increasingly utilized [5,8].

The landscape of rescue therapies for NMIUC has seen large developments in recent decades [6]. For over 20 years, valrubicin was the only FDA-approved intravesical rescue therapy for patients with BCG-unresponsive CIS despite an unfavorable disease-free rate of only 10% at 1-year post-treatment. Subsequently, pembrolizumab and nadofaragene firadenovec were approved for BCG-unresponsive CIS given the attractive initial response rates, but after a year of follow-up, the disease-free rate dropped to 19% and 24%, respectively [12,16]. Beyond the expanding list of bladder cancer treatment modalities, the use of novel combinatory intravesical therapy is a promising alternative [17,18,19]. Sequential intravesical gemcitabine and docetaxel (Gem/Doce) were initially utilized as a rescue intravesical therapy following BCG failure and has subsequently demonstrated success across a number of disease states [4,14,20,21]. Despite an increase in effective alternative therapies, tumor relapse does occur. Within this population, subsequent recurrences may be due in part to occult disease, unreachable by conventional intravesical therapies (i.e., upper urinary tracts and prostatic urethra) or the upregulation of innate cellular defense mechanisms against toxic metabolites (multidrug resistance-associated protein and p-glycoprotein efflux pumps).

In vitro studies have demonstrated an exfoliating effect of cationic chemotherapeutics, augmenting the delivery of subsequent taxane agents and providing improved drug penetration to the urothelium [22]. However, multiple trials of taxane-containing therapies such as docetaxel may increase cellular resistance mechanisms. Cabazitaxel is a taxane with anti-tumor activity and potency similar to that of docetaxel but with lower affinity to the multidrug resistance 1 protein [23]. A phase III clinical trial found that use of cabazitaxel plus prednisone improved overall survival for patients with metastatic castration-resistant prostate cancer whose disease had progressed during or after docetaxel-based therapy [24]. Recently, a phase I trial reported the usage of cabazitaxel in combination with gemcitabine and cisplatin (CGC) in a high-risk cohort of 19 patients with heavily pretreated, BCG-unresponsive NMIBC [25]. Here, complete response rate was 89% with an estimated recurrence-free survival of 64% at 2 years [25]. In a comparable population of patients with heavily pretreated NMIUC, we found GCP performed similarly with a 2-year RFS of 52%.

After developing the occult disease of the upper tract and/or prostatic urethra/ducts, patients are unlikely to respond to intravesical therapy alone. Prior studies suggest high rates of extravesical disease recurrence after intravesical therapy, particularly in patients with a history of multiple prior regimens. In a cohort of 307 patients with NMIBC, Herr reported that 78 (25%) eventually developed upper tract disease and 61 (24%) developed prostatic urethral disease. In the phase I trial investigating salvage CGC, 3/18 (17%) patients relapsed solely within the prostatic urethra [25]. This is the first time there has been equal efforts in treating extravesical disease as opposed to strictly treating disease of the bladder. Most clinical trials exploring bladder cancer therapies today exclude patients with extravesical disease. However, clearly there is a rising incidence of such occurrences with a recent study citing a median time for the development of urethral and/or upper tract disease of 3.5 years for patients with high-risk NMIBC failing two or more courses of BCG [26]. As patients pursue more rescue therapies, there will be a greater need for effective treatments in this population of patients prone to extravesical relapse. GCP provides one means for cancer control in this population.

There are limitations to this study. Although a prospective clinical trial would allow for the rigorous collection of outcomes, the retrospective method of data collection may concede selection bias. Given that this was a single-arm investigation, the individual value of each agent included in the GCP regimen cannot be derived here. While treating positive HG urine cytology is controversial, evidence suggests barbotage cytology is as reliable as biopsy histopathology [27]. Variable AE reporting standards throughout the study period and within the medical record may misrepresent the true toxicity profile of GCP therapy. Lastly, the results presented here are derived from a single institution with rigorous NMIBC surveillance and treatment protocols, which may not be fully replicated in lower volume centers.

## 5. Conclusions

In this cohort of patients with high-risk non-muscle invasive urothelial carcinoma, sequential endoluminal gemcitabine and cabazitaxel with concurrent intravenous pembrolizumab rescued a substantial number of patients following multiple prior treatment failures. While the risk of progression is a concern in this population, there was a low rate of cancer-specific mortality.

## Figures and Tables

**Figure 1 cancers-16-02561-f001:**
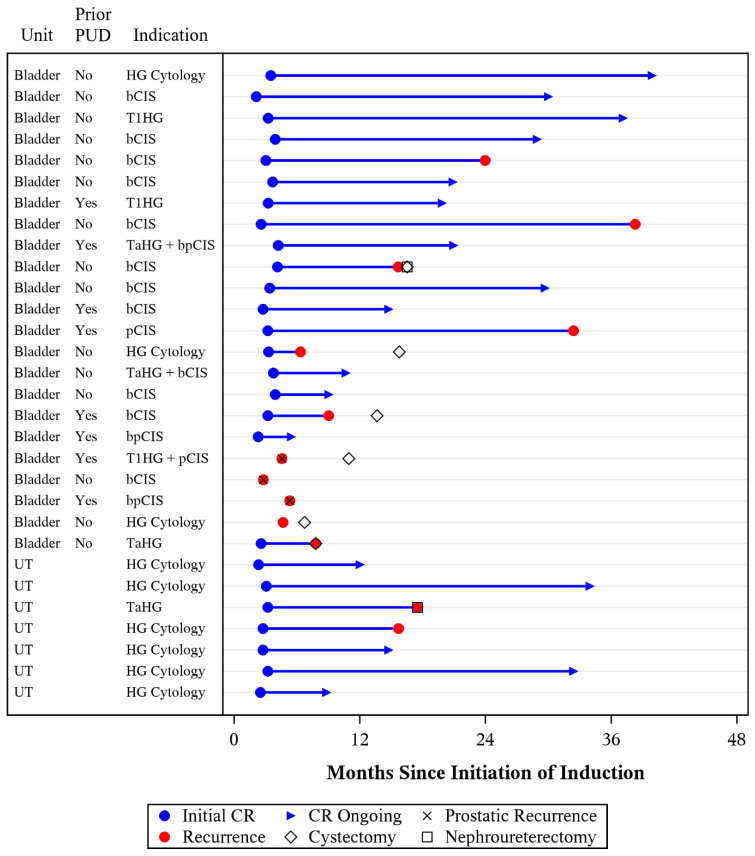
Clinical outcomes of individual units receiving GCP.

**Figure 2 cancers-16-02561-f002:**
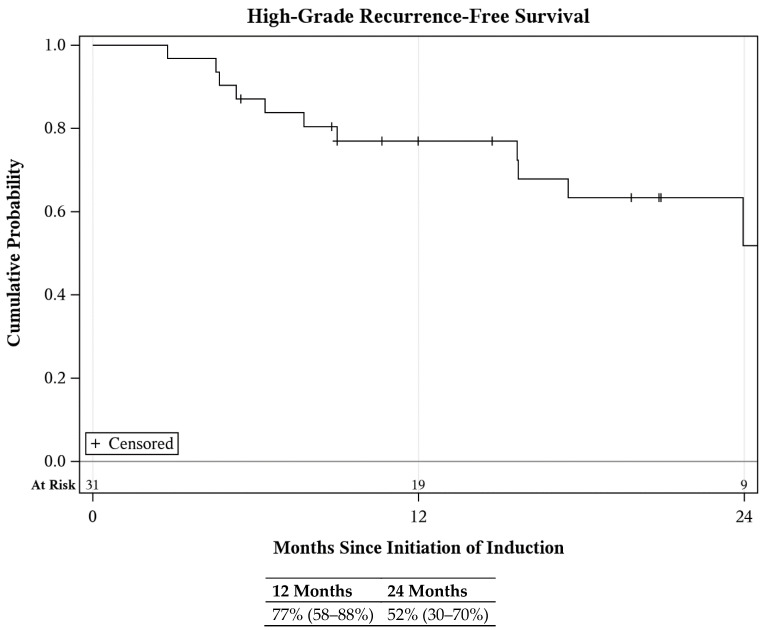
Recurrence-free survival among all treated units following GCP treatment.

**Figure 3 cancers-16-02561-f003:**
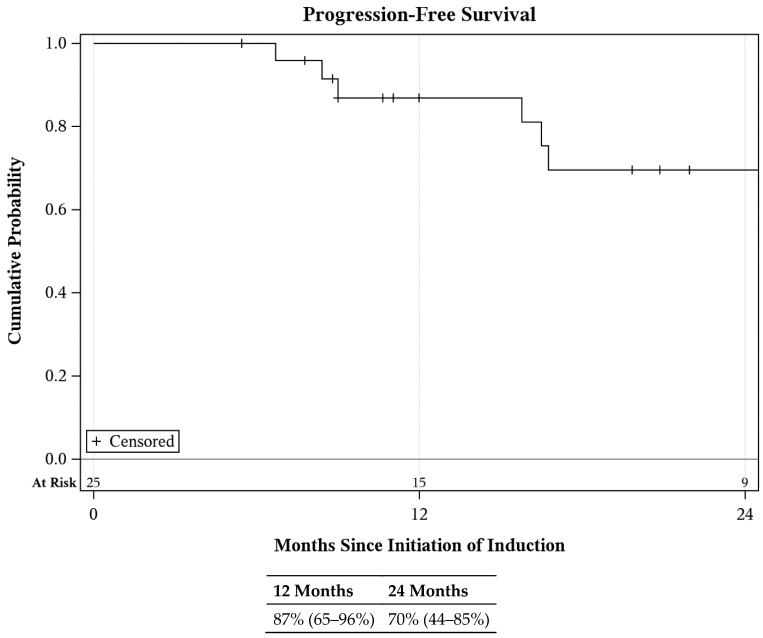
Progression-free survival following GCP treatment.

**Table 1 cancers-16-02561-t001:** Clinicopathological characteristics prior to GCP treatment.

		Urinary Tract Location	
	Total	Lower	Upper
No. of Patients (%)	26 (100)	24 (92)	6 (23)
Median Age (IQR)	73 (66–80)	72 (66–80)	75 (71–80)
Sex (%)			
Male	25 (96)	23 (96)	6 (100)
Female	1 (3.8)	1 (4.2)	-
Race (%)			
White, Non-Hispanic	26 (100)	24 (100)	6 (100)
No. of Treated Units (%)	31 (100)	24 (77)	7 (23)
Pretreatment Indication (%) **			
bCIS	12 (39)	12 (50)	-
bpCIS	2 (6.5)	2 (8.3)	-
pCIS	1 (3.2)	1 (4.2)	-
T1HG	2 (6.5)	2 (8.3)	-
TaHG	2 (6.5)	1 (4.2)	1 (14)
T1HG + pCIS	1 (3.2)	1 (4.2)	-
TaHG + bCIS	1 (3.2)	1 (4.2)	-
TaHG + bpCIS	1 (3.2)	1 (4.2)	-
High-Grade Cytology	9 (29)	3 (13)	6 (86)
Prior pCIS (%)	-	8 (36)	-
No. of Prior Inductions (%)			
0	2 (6.5)	-	2 (29)
1–2	5 (16)	1 (4.2)	4 (57)
3–4	17 (55)	16 (67)	1 (14)
5–6	7 (23)	7 (29)	-
Median No. of Prior Inductions (IQR)	4 (3–4)	4 (3–5)	1 (0–1)
Previously Received Treatments (%)			
BCG	22 (71)	21 (88)	1 (14)
Docetaxel-Containing Regimen *	29 (94)	24 (100)	5 (71)
Median No. of Prior Docetaxel Instillations (IQR)	27 (8–47)	39 (21–54)	4 (0–8)

* Docetaxel-containing regimens included doxorubicin–docetaxel, gemcitabine–docetaxel, valrubicin–docetaxel, Quadruple Chemotherapy (gemcitabine, doxorubicin, mitomycin-C, docetaxel), and single-agent docetaxel. ** bCIS, pCIS, and bpCIS represent carcinoma in situ of the bladder, prostate, and both bladder and prostate, respectively.

**Table 2 cancers-16-02561-t002:** Study treatment characteristics.

		Urinary Tract Location	
	Total	Lower	Upper
No. of Patients (%)	26 (100)	24 (92)	6 (23)
Received Pembrolizumab (%)			
Yes	24 (92)	22 (92)	6 (100)
No	2 (7.7)	2 (8.3)	0
Median # of Pembro doses (IQR)	10 (5–16)	11 (5–15)	10 (10–20)
No. of Treated Units (%)	31 (100)	24 (77)	7 (23)
Received Gem/Cabaz Maintenance (%)			
Yes	30 (97)	23 (96)	7 (100)
No	1 (3.2)	1 (4.2)	-
Median # of Maintenance Instillations (IQR)	7 (3–13)	11 (3–16)	6 (5–8)

**Table 3 cancers-16-02561-t003:** Adverse events following Gem/Cabaz with Pembrolizumab therapy.

	Grade 1	Grade 2	Grade 3	Total
Total Reported Adverse Events	29	33	5	67
Gem/Cabaz (*n* = 26)	25	25	2	52
UTI	-	8 (31)	1 (3.8)	9 (35)
Bladder Spasm	-	7 (27)	-	7 (27)
Abdominal/Flank Pain	3 (12)	4 (15)	-	7 (27)
Hematuria	4 (15)	2 (7.7)	-	6 (23)
Dysuria	5 (19)	-	-	5 (19)
Frequency/Urgency	4 (15)	-	-	4 (15)
Fatigue	3 (12)	-	-	3 (12)
Nausea/Vomiting	2 (7.7)	1 (3.8)	-	3 (12)
Rash	1 (3.8)	1 (3.8)	-	2 (7.7)
Retention	1 (4.3)	-	-	1 (3.8)
Anxiety	-	1 (3.8)	-	1 (3.8)
Dry Mouth	1 (3.8)	-	-	1 (3.8)
Stone Formation	-	-	1 (3.8)	1 (3.8)
Ureteral Stricture	-	1 (3.8)	-	1 (3.8)
Hair Loss	1 (3.8)	-	-	1 (3.8)
Pembrolizumab (*n* = 24)	4	8	3	15
Rash	-	3 (13)	-	3 (13)
Transaminitis	1 (4.2)	1 (4.2)		2 (8.3)
Arthralgias	-	2 (8.3)	-	2 (8.3)
Flu-like Symptoms	1 (4.2)	-	-	1 (4.2)
Pruritis	1 (4.2)	1 (4.2)	-	2 (8.3)
Pneumatosis Intestinalis	-	-	1 (4.2)	1 (4.2)
Constipation	1 (4.2)	-	-	1 (4.2)
Guttate Psoriasis	-	1 (4.2)	-	1 (4.2)
Pancreatitis	-	-	1 (4.2)	1 (4.2)
Capillary Leak Syndrome	-	-	1 (4.2)	1 (4.2)
Patients Reporting Adverse Events (%)	22 (85)			

Adverse events (AEs) are assigned according to CTCAE V5. Percentages (in parentheses) are based upon the number of patients receiving each treatment (Gem/Cabaz or pembrolizumab).

## Data Availability

Data included in this study are not publicly available.

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
