# Peer review of "Sequential Endoluminal Gemcitabine and Cabazitaxel with Intravenous Pembrolizumab as a Bladder-Preserving Strategy for Docetaxel-Unresponsive Non-Muscle Invasive Urothelial Carcinoma Following Transurethral Resection of Bladder Tumor"

_cancers, 2024, doi:10.3390/cancers16142561_

Round 1
Reviewer 1 Report
Comments and Suggestions for Authors
This is an interesting paper reporting the potential value of a sequential induction based on intravesical G+C plus intravenous P, in Docetaxel unresponsive high-risk NMICC, in the context of bladder preservation strategy. The reported data of 52% patients remaining disease-free at 2 years follow-up and a low rate of adverse events of 19% support the proposed combination therapy as appropriated scheme of bladder preservation in docetaxel unresponsive patients.
Potential improvement:
1. Since patients were treated with GCP and complete TURBT, I would suggest adding this in the title to make the paper of more interest…Probably, adding bladder preservation strategy would be of interest too for a modified title.
2. Is the following paragraph in Mat&Met “Patients were excluded from efficacy analysis if they did not have pathological high-grade disease or a definitively positive localizing high-grade cytology and/or did not undergo follow-up surveillance” related to exclusion criteria? If so I found a bit unclear the paragraph and therefore I suggest rewording for clarity. Also using these criteria how many cases were excluded from the analysis?
3. In surveillance, the authors have used cytology, biopsies and FISH analysis. Is all this necessary for follow up? What is the recommended protocol by the authors regarding cytology? Biopsy? Combination of both? Fish when?
4. All patients that developed progression were offered directly cystectomy? Or were offered radiation therapy as part of bladder preservation strategy? It would be of interest having the experience of the authors.
5. Would it be Ok is the authors add, as foot note in table 1 the meaning of bCIS, pCIS or bpCIS.
Reviewer 2 Report
Comments and Suggestions for Authors
Nothing to suggest for the Introduction, M&M, Results, and Discussion sections.
Comment for the author:
Very promising results regarding 2-year survival and progression outcomes with high patient tolerance and compliance. It would be interesting to read about long-term survival outcomes and progression. It might be challenging to try to increase the dosage schedule, as you had few and mostly low-grade AEs. For sure, this regimen is another solution in the landscape of salvage therapies for NMIUC. Congratulations on the nice work. Eager to know the longer-term duration of response.
